# Antibacterial Activity of Silver-Modified CuO Nanoparticle-Coated Masks

**DOI:** 10.3390/bioengineering11121234

**Published:** 2024-12-05

**Authors:** Tanuja Udawant, Prajkta Thorat, Payal Thapa, Manali Patel, Saroj Shekhawat, Roshni Patel, Ankit Sudhir, Om Hudka, Indra Neel Pulidindi, Archana Deokar

**Affiliations:** 1Modern College of Arts, Science, and Commerce, Pune 411005, India; tanujaudawant159@gmail.com (T.U.); prajaktathorat03@gmail.com (P.T.); 2Department of Life Sciences, School of Science, Gujarat State Fertilizers and Chemicals University, Vadodara 391750, India; 23msc02019@gsfcuniversity.ac.in (P.T.); 23msc02005@gsfcuniversity.ac.in (M.P.); saroj.shekhawat@gsfcuniversity.ac.in (S.S.); roshni.patel@gsfcuniversity.ac.in (R.P.); ankit.sudhir@gsfcuniversity.ac.in (A.S.); 3Department of Chemical Sciences, School of Science, Gujarat State Fertilizers and Chemicals University, Vadodara 391750, India; 22msc01014@gsfcuniversity.ac.in; 4Jesus’ Scientific Consultancy for Industrial and Academic Research (JSCIAR), Tharamani 600113, India

**Keywords:** Ag-CuO, Ag-ZnO, nanoparticles, sonication, airborne bacteria, masks, Zone of inhibition, ZoI, reactive oxygen species, ROS

## Abstract

A green and cost-effective sonochemical synthetic method was followed for coating silver-modified copper oxide (Ag-CuO) nanoparticles (NPs) on disposable surgical mask. The NP-coated masks were systematically characterized using XRD and FT-IR for understanding the structural and surface functionalities. In addition, the field emission scanning electron microscopy (FE-SEM) analysis showed the homogeneous coating of Ag-CuO NPs over the mask fibers. The average particle size of Ag-CuO was found to be ~70 nm. The NP-coated masks are useful to combat a broad range of bacterial species by taking the unique advantage of the synergistic effect of Ag and metal oxide (CuO and ZnO) NPs for the generation of reactive oxygen species (ROS). Zone of inhibition (ZoI) studies demonstrated antibacterial activity against both Gram-positive *S. aureus* and Gram-negative *E. coli* bacteria, probably due to the elevated production of ROS by the defect structure of the Ag-modified metal oxide NPs. The material was found to be effective against both airborne and soil-borne bacteria. We repeat that this paper deals only with the killing effect of the nanoparticles (Ag-modified CuO) on bacteria, and no studies on viral species are performed.

## 1. Introduction

Owing to the worldwide high mortality rate, each day due to transmissible respiratory diseases, there is an urgent need for designing masks capable of capturing and inactivating microbes within a short period of time [1,2,3,4]. The sonochemical method is a potential technique for the in situ synthesis and coating of modified metal oxides in nano dimensions on various substrates [5,6,7]. This method has been proven to be green and cost-effective for several reasons, like operation at lower temperatures and pressures, reduction in energy consumption compared to some high-temperature methods, minimized use of hazardous or toxic chemicals, and the formation of fewer byproducts, as well as a decrease in the need for extensive post-synthesis purification steps [8,9,10]. Moreover, it can be scaled up for industrial applications. 

Previously we reported on an ultrasound-assisted coating of copper oxide (CuO) and zinc oxide (ZnO) NPs on bandages and cotton fabrics. This technique was proven to be effective against multi-drug-resistant (MDR) bacterial strains [7,10]. The primary mechanism of antimicrobial activity is via the production of reactive oxygen species (ROS). Metal oxide NPs attach to microbes, which in turn provokes the enhancement of the intracellular oxidative stress. Moreover, previous studies demonstrated that modified CuO NPs show superior antimicrobial activity due to the elevated ROS production [11,12]. The mechanism of the antibacterial activity of silver NPs includes the release of silver ions, the generation of ROS, interactions with proteins, damaging the DNA via binding to the DNA, and also inhibiting the enzyme activity [13,14,15,16]. These diverse processes collectively contribute to the antibacterial efficacy of Ag NPs. 

With this motivation to enhance protective measures against microbial infections, especially bacteria, efforts were devoted towards the ultrasound-assisted coating of surgical masks with either Ag-modified CuO or ZnO NPs for combating bacterial infections (Figure 1). 

As-synthesized powders of NPs were subjected to XRD analysis to gain insights into the defects present in both Ag-CuO (AC) and Ag-ZnO (AZ) NPs. Electron paramagnetic resonance (EPR) analysis has confirmed the generation of elevated ROS in Ag-CuO NPs, indicating the superior antibacterial activity. An intensified efficacy in the eradication of both Gram-positive *S. aureus* and Gram-negative *E. coli* bacteria with Ag-CuO NPs was observed. This enhanced rate of bacterial elimination is likely a result of an increased production of ROS.

## 2. Materials and Methods

### 2.1. Synthesis and Deposition/Coating of Ag-CuO (AC) and Ag-ZnO (AZ) NPs on Masks

All the chemicals used in the study were procured from Sigma Aldrich Powai, Mumbai, India. The coating of Ag-CuO/ZnO NPs on masks was performed using a bath sonicator (40 KHz). The surface of the mask to be coated (Modenna disposable surgical face mask made up of polypropylene, PP) was attached face down to the substrate in such a way that only one side of the mask was being coated with NPs. AgNO_3_ and copper/zinc acetate (0.01 M) (silver nitrate and copper/zinc acetate 3:1 mol/mol) were dissolved in 100 mL ethanol–water in a ratio of 9:1 (*v*/*v*). The solutions were heated up to 63 °C followed by the addition of 7–10 mL of an aqueous solution of NH_4_OH (28–30 %) into the reaction medium to adjust the pH to ~8. At the end of the reaction, the color of the solution was changed from white to dark brown in the case of CuO NPs. A light brown color was observed in the case of ZnO NPs. The coated masks were cleaned once with ethanol, and then dried in an air-oven at 50 °C. The coating of NPs was continued for another 35 min. The coated masks were cleaned thoroughly with ethanol, and then dried in an air-oven at 50 °C. The settled NPs’ precipitate was washed by centrifugation, dried, and subsequently subjected to further characterization [17].

### 2.2. Characterization of Ag-CuO/ZnO

NPs coated masks and powders: Dried powders obtained from coated masks were subjected to XRD analysis (Bruker D8 Advance X-ray powder diffractometer with Cu Kα as X-ray source, Bruker, Karlsruhe, Germany) and FT-IR analysis (Shimadzu, ATR-FT-IR, Diamond detector, Tensor II, Tokyo, Japan). Dried Ag-CuO/ZnO NPs coated masks were coated with conducting gold and subjected to FE-SEM (FEI Nova NanoSEM 450 with Energy dispersive spectroscopy (EDS): Bruker XFlash 6130) to confirm the uniform distribution of NPs onto the masks as well as to observe the morphology of the NPs.

### 2.3. EPR Study

The production of ROS was determined by EPR spectroscopy using a Bruker EPR 100d X- band spectrometer (Bruker, Karlsruhe, Germany). 5,5-dimethyl-1-pyrroline N-oxide (DMPO) was used as a spin trap and DMSO was used as the hydroxyl radical (OH^.^) scavenger. The aqueous suspensions of the NPs were added to the DMPO and the samples were drawn by a syringe into a gas-permeable Teflon Capillary of 0.082 cm inner diameter, 0.038 inch wall thickness, and 15 cm length. The capillary was folded twice and inserted into a narrow quartz tube opened at both ends. This tube was placed in the EPR cavity, and measurements were carried out under the following conditions: frequency, 9.74 GHz; microwave power, 20 mW; scan width, 65 G; resolution, 1024; receiver gain, 2 × 10^5^; (conversion time, 82 ms); time constant, 655 ms; sweep time, 84 s; scans, 2; modulation frequency, 100 kHz. After the acquisition of the data, the spectra were processed using the Bruker WIN-EPR software version 2.11 for baseline correction [18].

### 2.4. Antimicrobial Activity Test

Antibacterial activities of synthesized AC and AZ NPs were examined against the soil and air bacteria. Antibacterial activity was tested upon both Gram-positive, *S. aureus*, and Gram-negative, *E. coli*, bacteria. 

#### 2.4.1. Culturing Soil Bacteria

The LB agar plates were used for culturing soil bacteria. The Luria–Bertani (LB agar) medium was prepared and autoclaved at 121 °C/15 psi, and afterwards, the culture plates were poured with and without the addition of NPs into the LB agar and solidified to test the antibacterial properties. The LB plate without NPs was used as control and the other experimental plates were prepared with a 1000, 100, and 10 ppm concentration of the NPs (AC/AZ) in the LB agar to test the dose-dependent antibacterial effect. Random soil samples were collected from a near by hospital premises. Then, 1 g soil sample was mixed with 10 mL LB broth. The pour plate method was used to culture soil microorganisms in the control and NPs containing culture plates. The plates were incubated at 37 °C for 24 h in the incubator.

#### 2.4.2. Culturing Air Bacteria

The LB agar plate was used for culturing air microorganisms as discussed above. To capture air microorganisms, the plates were kept for open incubation in the laboratory for 10 min and then transferred to the incubator and kept for incubation for 24 h.

#### 2.4.3. Agar-Plate Method

The antimicrobial efficacy of AC and AZ against *E. coli* was also evaluated. For this test, *E. coli* was cultured in nutrient agar (NA) medium at 37 °C. The colonies were then inoculated into nutrient broth (NB) and incubated at 37 °C for 24 h. To assess the antimicrobial activity of AC and AZ, nanoparticle solutions of different concentrations (10, 100, 1000, and 2000 ppm) were prepared in sterile distilled water. These solutions were added to a 2 % nutrient agar, which was then poured into Petri dishes. Once solidified, 100 µL of the *E. coli* culture was evenly spread on the plates containing nanoparticle-infused agar. The plates, modified with varying concentrations of AC and AZ NPs and infused with *E. coli*, were incubated at 37 °C for 24 h. The antibacterial effect of each nanoparticle concentration was determined by measuring the diameter of the zone of inhibition (ZoI), a reflection of antibacterial activity.

#### 2.4.4. ZoI Method

The antibacterial studies of AC and AZ NP powders from masks were evaluated using the ZoI method against Gram-positive, *S. aureus*, and Gram-negative, *E. coli*. Overnight cultured bacterial strains were transferred into a fresh LB medium and agitated for 4 h at 37 °C. Furthermore, 108 cell cultures were harvested by centrifugation and washed with fresh 0.9 % sodium chloride (NaCl) solution. The strains were uniformly swabbed onto individual plates using sterile cotton swabs. Subsequently, 20 µL of the nanoparticle solution sample was applied to the center of each well on all plates, using a sterile micropipette. Following an incubation period at 37 °C for 24 h, the varying levels of the ZoI were assessed using the Hi antibiotic zone scale. 

## 3. Results

### 3.1. Morphology and Size of AC NPs Coated Disposable Surgical Masks

Figure 2 represents the FE-SEM images of the Ag-CuO NP-coated surgical mask. FE-SEM was carried out to observe the morphology and the resultant coating density of NPs on the mask. A homogenous and dense coating of NPs was observed throughout the mask fibers. Few fibers demonstrate aggregated forms of NPs (Figure 2b,c, magnified view of Figure 2a). Coated Ag-CuO NPs were spherical in shape, with an average particle size of ~70 nm (Figure 2d). The loadings of CuO and ZnO NPs are around ~1 wt. %, and such a loading is dependent on the appropriate choice of the precursor concentration [19]. 

### 3.2. Structural Characterization of AC and AZ NPs Using XRD Analysis

XRD patterns of CuO NPs and Ag-CuO NPs synthesized in ethanol–water media by the sonochemical method are shown in Figure 3. The XRD pattern of AC NPs (red line) produced after the sonochemical reaction of a mixture of AgNO_3_ and copper acetate in an appropriate molar ratio (3:1) is shown in Figure 3. For comparison, the XRD pattern of sonochemically synthesized CuO NPs (black line) is also shown in Figure 3.

The peaks at 2θ (in degrees) values of 32.52, 35.52, 38.76, 48.88, 58.50, and 61.64 were assigned to (110), (002), (111), (202), (202), and (113) planes of monoclinic CuO NPs, respectively. In contrast to the XRD pattern of CuO, in the case of Ag-CuO systems, the peaks typical of Ag NPs predominated over the CuO signals as expected, owing to the higher amounts of Ag^+^ precursor compared to the Cu^2+^ (Figure 3). However, the peak positions of the monoclinic CuO remained unaltered, indicating that there is no doping of Ag into the CuO lattice. In AC NPs, four characteristic peaks corresponding to the Ag NPs were also observed at 2θ values of 37.98°, 44.05°, 64.22°, and 76.79°, corresponding to the planes indexed as (111), (200), (202), and (311), respectively, characteristic of the cubic crystal system (a = 4.1090 Å). The results are consistent with the report from Shankar and co-workers (2004) [20]. Therefore, the formation of crystalline NPs of Ag-modified CuO as a consequence of the green synthesis method adopted can be confirmed. As Ag could be observed to represent a significant peak area of the Ag modified CuO NPs, it is understood that some silver has occupied the interstitial sites of the CuO lattice without causing any other defects [21].

The XRD patterns of ZnO NPs and Ag-modified ZnO (AZ) NPs synthesized in ethanol (et) –water mixture by the sonochemical method are shown in Figure 4 and Figure 5, respectively. The XRD pattern of ZnO NPs showed peaks typical of the Wurtzite phase (Figure 4, JCPDS No. 89–1397). The XRD pattern of Ag-ZnO NPs showed peaks at 31.84°, 34.38°, 36.22°, 47.50°, 56.60°, 62.80°, and 67.89° that were, respectively, indexed to the (100), (002), (101), (102), (110), (200), and (112) planes, corresponding to the Wurtzite ZnO structure (JCPDS No. 89-1397). In addition, planes typical of Ag were observed at the 2θ value at 38.58°*, 44.96°*, 64.99°*, and 77.74°* Ag (*) and have been indexed to the planes (111), (200), (220), and (311), respectively. These values are characteristic of the cubic Ag structure (JCPDS No. 89-3722). The Ag-modified ZnO NPs consist of both the Wurtzite phase of ZnO and the cubic phase of Ag NPs (Figure 5). These results further confirm that ZnO/Ag nanocomposites were made up of the individual ZnO (Wurtzite) and Ag (cubic) phases (Figure 5). 

### 3.3. Surface Functionality of Ag Modified CuO and Ag Modified ZnO NPs Using FT-IR Analysis

FT-IR spectroscopic analysis proved the synthesis of the Ag-CuO NPs in ethanol–water by a mild sonication method. The FT-IR spectra of CuO and Ag-modified CuO (AC) NPs were shown in Figure 6 and Figure 7, respectively. Typical bands observed include O-H stretching (a broad band at 3000–3700 cm^−1^) centered at about 3367 cm^−1^, and the O-Cu-O bond stretching (674 cm^−1^), sp^3^ Cu-O bond (1033 cm^−1^), sp^2^ Cu-O bond (1662 cm^−1^), and asymmetric Cu-O (1341 cm^−1^) bond may be attributed to the metal precursors utilized for the synthesis of CuO (Figure 6) and AC NPs (Figure 7). Stretching frequency at 588 cm^-1^ is direct evidence of the formation of AC NPs (Figure 7). The FT-IR spectral features support the XRD result that Ag is not doped into the CuO lattice, but that Ag nanoparticles as well as CuO NPs exist as two individual phases in the AC NPs. 

FT-IR spectra of ZnO and Ag-modified ZnO (AZ) NPs are shown in Figure 8. Transmission bands typical of O-H stretching (a broad band at 3000–3700 cm^−1^) were centered at about 3367 cm^−1^. Bands typical of O-Zn-O bonds (703 cm^−1^), sp^3^ Zn-O bonds (1031 cm^−1^), sp^2^ Zn-O bonds (1553 cm^−1^), asymmetric Zn-O bond (1384 cm^−1^), and symmetric Zn-O bond stretching vibration (1496 cm^−1^) were observed. The stretching frequency at 595 cm^−1^ is a direct evidence of the formation of ZnO NPs. The FT-IR spectrum of Ag-ZnO NPs showed bands for O-H stretching (a broad band at 3000–3700 cm^−1^) centered at about 3365 cm^−1^ and O-Zn-O bonds (705 cm^−1^), sp^3^ Zn-O bonds (1031 cm^−1^), sp^2^ Zn-O bonds (1554 cm^−1^), asymmetric Zn-O bonds (1391 cm^−1^), and symmetric C-O bonds (1493 cm^−1^). The stretching frequency at 592 cm^−1^ is a direct evidence of the formation of Ag-modified ZnO NPs, and the results are in concurrence with the XRD analysis results, which show that Ag is not doped into the ZnO lattice, whereas two individual phases of ZnO (Wurtzite) and Ag (cubic) were present in the Ag-modified ZnO system. 

### 3.4. Evaluation of Antibacterial Activity

The ROS production ability of NPs was determined by the EPR spin trapping technique using the famous spin trap, namely DMPO. Four well-resolved peaks generated from the DMPO-OH adduct demonstrate the presence of hydroxyl radicals (OH^.^) in the NP suspensions (Figure 9) [18]. 

Effect of solvent: The synthesis of metal oxide (either CuO or ZnO) NPs was carried out using two distinct solvents to investigate their impact on the generation ROS with the aim of enhancing the efficacy in bacterial eradication. Findings indicated that CuO NPs synthesized in ethanol resulted in more intense EPR peaks compared to CuO NPs synthesized in water or ZnO NPs synthesized in either water or ethanol (Figure 9a,b). This difference could be probably due to smaller particle size of CuO NPs in ethanol medium [17]. Smaller particles possesses higher surface area and hence result in increased ROS production. ZnO NPs synthesized in water and ethanol demonstrated weak signal intensity indicating poor ability to generate hydroxyl radicals in the suspension. Based on these fundamental results, we have designed the Ag modified CuO that is surmised to have better antibacterial activity than the bare CuO and carried out antibacterial activity studies on Ag-CuO NPs alone (Figure 9b). In order to examine the nature of the ROS produced, we used DMSO to scavenge the hydroxyl radicals generated in NPs suspension. Figure 9c demonstrates that neither an obvious change in signal intensity of ZnO and CuO NPs nor an additional peak of DMPO-CH_3_ adduct was observed. The constant intensity even in the presence of the hydroxyl radical scavenger, namely, DMSO, of the quartet signal indicates the absence of hydroxyl radical and that the possible ROS could be either singlet oxygen or super oxide anion [12]. Furthermore, our previous study demonstrated that Zn doped CuO NPs, in addition to a major fraction of the hydroxyl radicals, could also produce singlet oxygen due to elevated number of defects in Zn doped CuO [15]. 

The preliminary screening of the nanomaterial coating’s antibacterial activity against soil- and air-exposed LB plates serves as a foundational step to demonstrate its broad-spectrum efficacy. This method simulates real-world contamination scenarios, validating the coating’s potential to inhibit diverse and undefined microbial populations. Establishing this general antimicrobial activity is critical before narrowing the focus to specific pathogenic strains like *E. coli* and *Streptococcus* (Appendix A). This approach ensures that the material is versatile and effective in varying environmental contexts, supporting its practical application as an antibacterial agent. Hence, herein, the antimicrobial activity of synthesized NPs, AC, and AZ was investigated against the soil- and airborne bacteria (Figure 10 and Figure 11). The obtained results showed that the prepared NPs exhibited antibacterial activities against both soil and air bacteria (Figure 10 and Figure 11). The NPs were active against the soil bacteria in a dose-dependent manner. AC showed more dose-dependent inhibition of microbial growth compared to AZ in the soil consortium at each checked concentration; however, the overall antimicrobial activity as compared to the control plate was prominent in both the cases.

Figure 10(1A,2A) show the bacterial lawn in the control plates. The dose-dependent effect of the NPs is evident in Figure 10(1B,2B) at the concentration of 1000 ppm. Antibacterial activity of the NPs at 100 and 10 ppm, respectively, are shown in Figure 10(1C,1D,2C,2D). The effect of NPs at lower concentrations was much less compared to the 1000 ppm concentration.

Figure 11(1A,2A) show the air bacteria in the control plates. The dose-dependent effect of the NPs is evident in Figure 11(1B) (500 ppm AC) and Figure 11(1C) (50 ppm AC); a similar dose-dependent observation is evident for AZ NPs as well. The effect of the 50 ppm concentration for both AC and AZ was less compared to 500 ppm as expected. However, it was significant when compared to the control plate [Figure 11(1A,2A)].

In contrast to the soil, the antimicrobial activity of AC/AZ was more pronounced in the air microorganisms (Figure 10 and Figure 11). AC NPs demonstrated more than 90 % inhibitory effects against air microflora at concentrations of 500 ppm. Both AC and AZ demonstrated the significant microbial inhibition of airborne microorganisms at a concentration of 50 ppm. (Figure 11). As evident from the results, both AC and AZ exhibited the highest microbial growth reduction in soil at a concentration of 1000 ppm and in the air at 500 ppm. It is clear that the synthesized AC NPs exhibited antimicrobial activity against both soil and air microorganisms and that the effect on the air microbes is more pronounced.

The *E. coli* culture exhibited resistance to AZ NPs at all the tested concentrations (10, 100, 1000, and 2000 ppm) (Figure 12A). In contrast, AC NPs demonstrated inhibitory effects against *E. coli* at relatively higher concentrations (1000 and 2000 ppm) compared to AZ (Figure 12B). Preethi et al. also demonstrated that AC NPs exhibit antibacterial activity against *E. coli* cultures at concentrations of 60, 80, and 100 ppm [22]. These NPs interact with essential components of bacterial cells, including the DNA, ribosomes, and lysosomes and contribute to oxidative stress, protein deactivation, alterations in cell membrane permeability, and changes in gene expression. The antibacterial activity of AC might be due to the production of ROS, such as hydroxyl radicals (OH^.^) (Figure 12), hydrogen peroxide (H_2_O_2_), or superoxide anions produced when NPs damage a bacterial cell wall, allowing NPs to enter the cell and preventing DNA replication, deactivating proteins, inhibiting enzymatic activity, and potentially causing electrolyte imbalance. Bacterial interior cellular component leakage will inevitably result in cell death [23]. 

Moreover, the stability of such sonochemical coating of ZnO and CuO NP onto bandages and other substrates has been proven [24]. It was proven that even after 65 washings, the NP-coated materials retain their antibacterial properties. In addition, after thorough washing and drying, the NPs do not tend to leach out of the substrate [25], ensuring the commercial utility of the material for practical applications. 

## 4. Summary and Conclusions

A mild sonochemical method was employed for the fabrication of Ag-modified CuO NP-coated surgical masks. Morphological analysis using FE-SEM revealed a uniform and compact coating of the modified metal oxide NPs across the fibers of the mask. This coating technique is not only effective but also cost-efficient, as it eliminates the need for any external binder during the coating process. An overall growth reduction of greater than 50% is observed for both air and soil bacteria for both Ag-modified CuO NPs (AC) and Ag-modified ZnO NPs (AZ). The highest growth reduction of ~75% was observed for the air consortium. The resulting coated masks exhibit significant potential antibacterial effects as a viable solution against a diverse array of bacterial and viral species, attributed to their remarkable ROS production capability. Thus, a potential marketable product, namely masks coated with Ag-modified CuO nanoparticles (NPs), is developed with antibacterial activity. Anti-viral studies on this material are yet to be performed. Any reference to viruses in this paper is intended to remind the readers of the harmful nature of these microbes in general as this generation has already experienced the disaster of COVID virus. Moreover, this method still requires experimental verification to demonstrate the harmlessness of these NP coatings in relation to the risks of their inhalation.

## Figures and Tables

**Figure 1 bioengineering-11-01234-f001:**
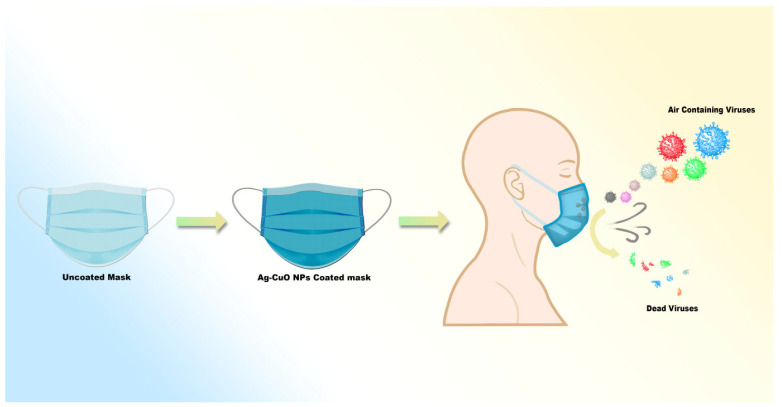
Schematic illustration of ultrasound-assisted coated surgical masks for the effective killing of bacteria.

**Figure 2 bioengineering-11-01234-f002:**
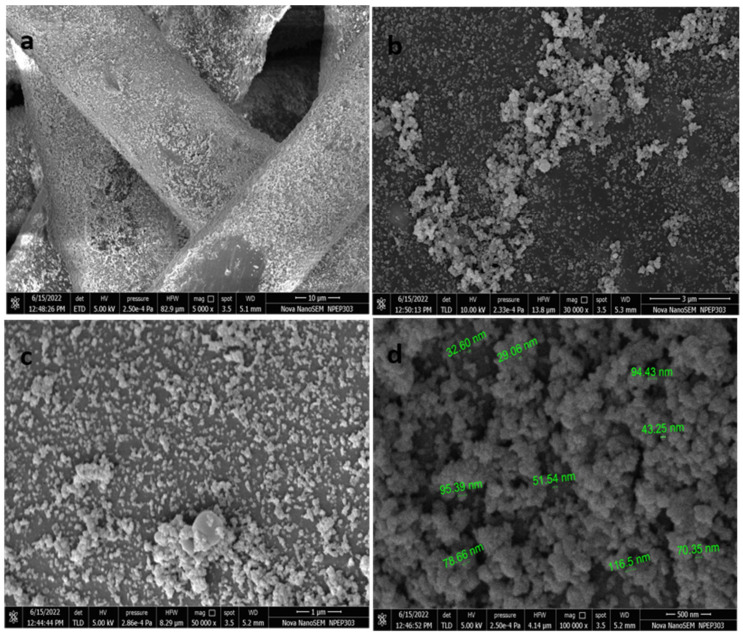
FE-SEM images of Ag-CuO (AC) NPs coated mask by mild sonication at different magnifications (**a**) 5000×, (**b**) 30,000×, (**c**) 50,000×, (**d**) 100,000×.

**Figure 3 bioengineering-11-01234-f003:**
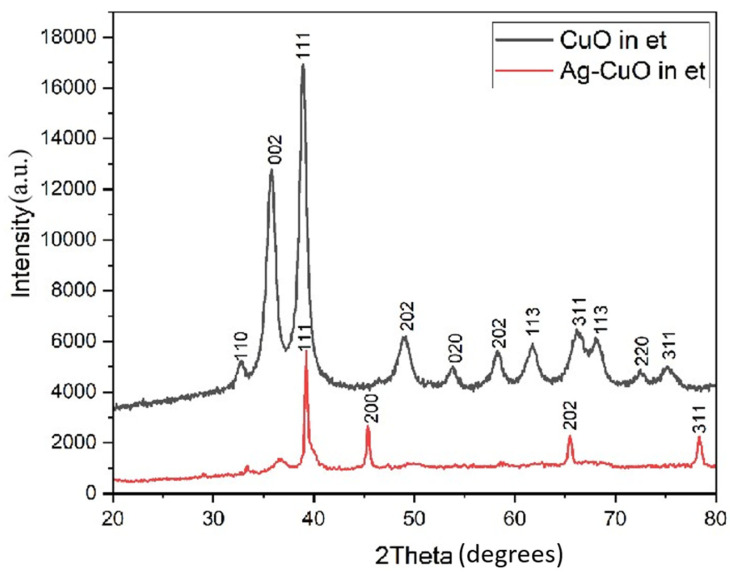
X-ray diffractograms of CuO and Ag-CuO NPs synthesized in ethanol (et) –water medium by mild sonication.

**Figure 4 bioengineering-11-01234-f004:**
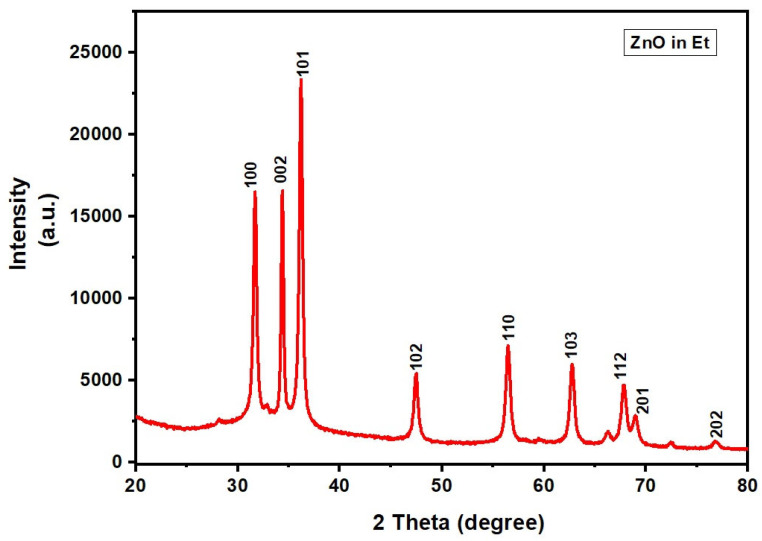
X-ray diffractogram of ZnO NPs synthesized in ethanol (Et)– water medium by mild sonication.

**Figure 5 bioengineering-11-01234-f005:**
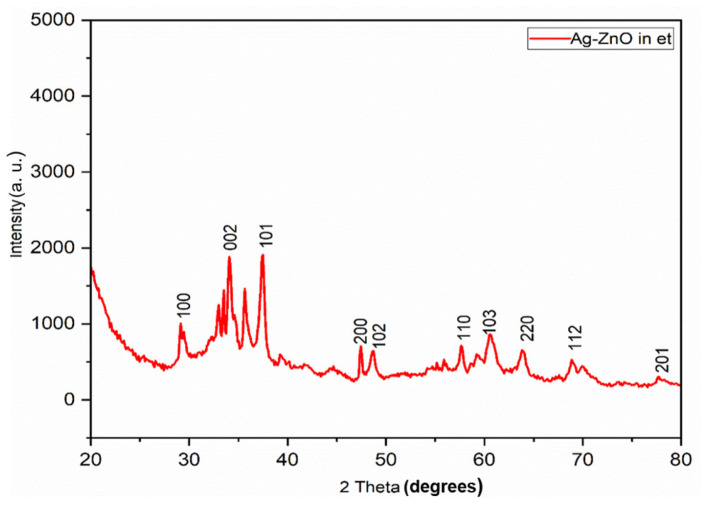
X-ray diffractogram of Ag-modified ZnO nanoparticles (NPs) synthesized in ethanol (et) – water medium by mild sonication.

**Figure 6 bioengineering-11-01234-f006:**
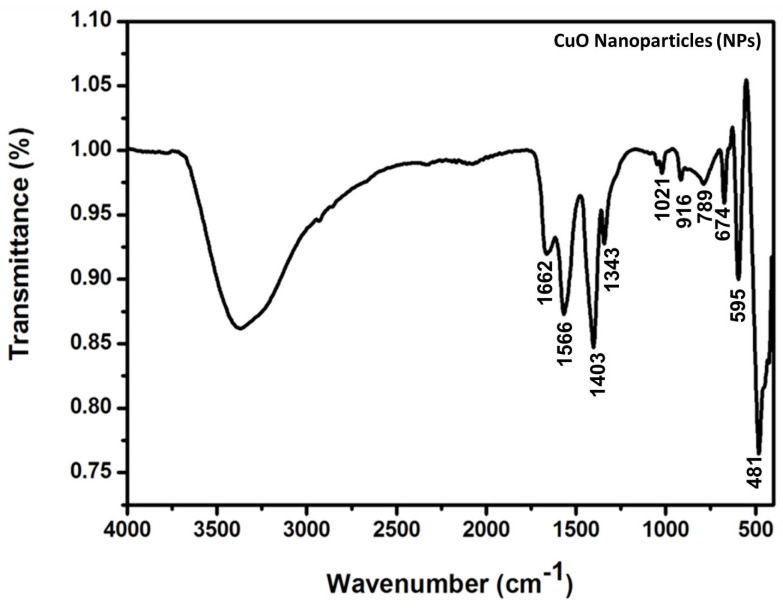
FT-IR spectrum of CuO NPs prepared in water medium by sonication method.

**Figure 7 bioengineering-11-01234-f007:**
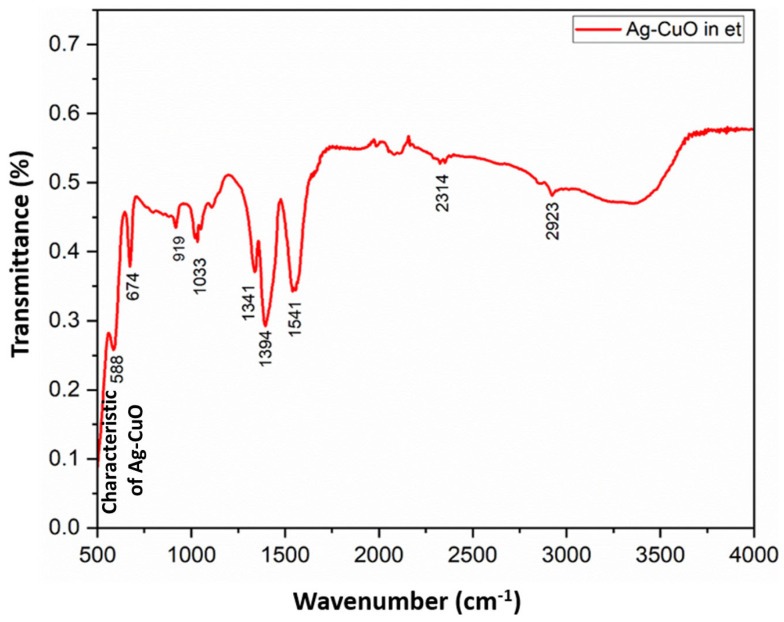
FT-IR spectrum of Ag-CuO NPs prepared in ethanol (et) –water medium by sonication method.

**Figure 8 bioengineering-11-01234-f008:**
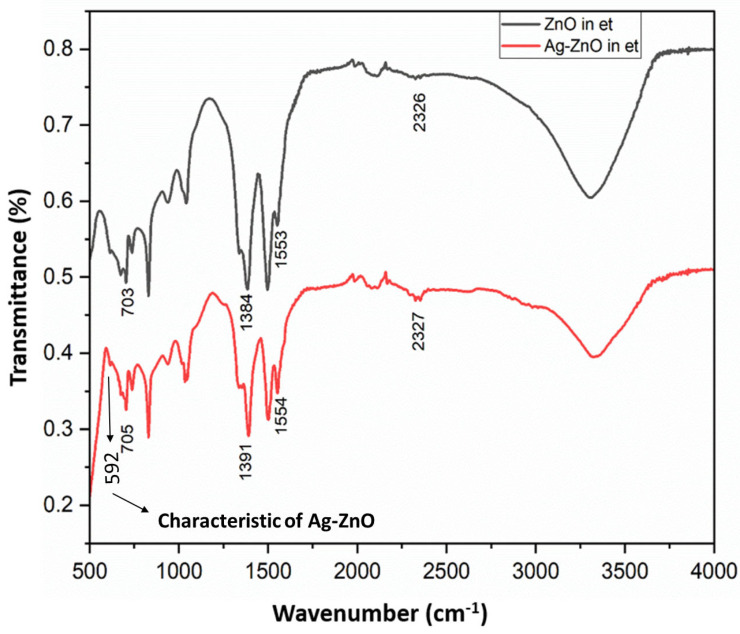
FT-IR spectrum of ZnO and Ag-ZnO NPs prepared in ethanol–water medium by mild sonication method.

**Figure 9 bioengineering-11-01234-f009:**
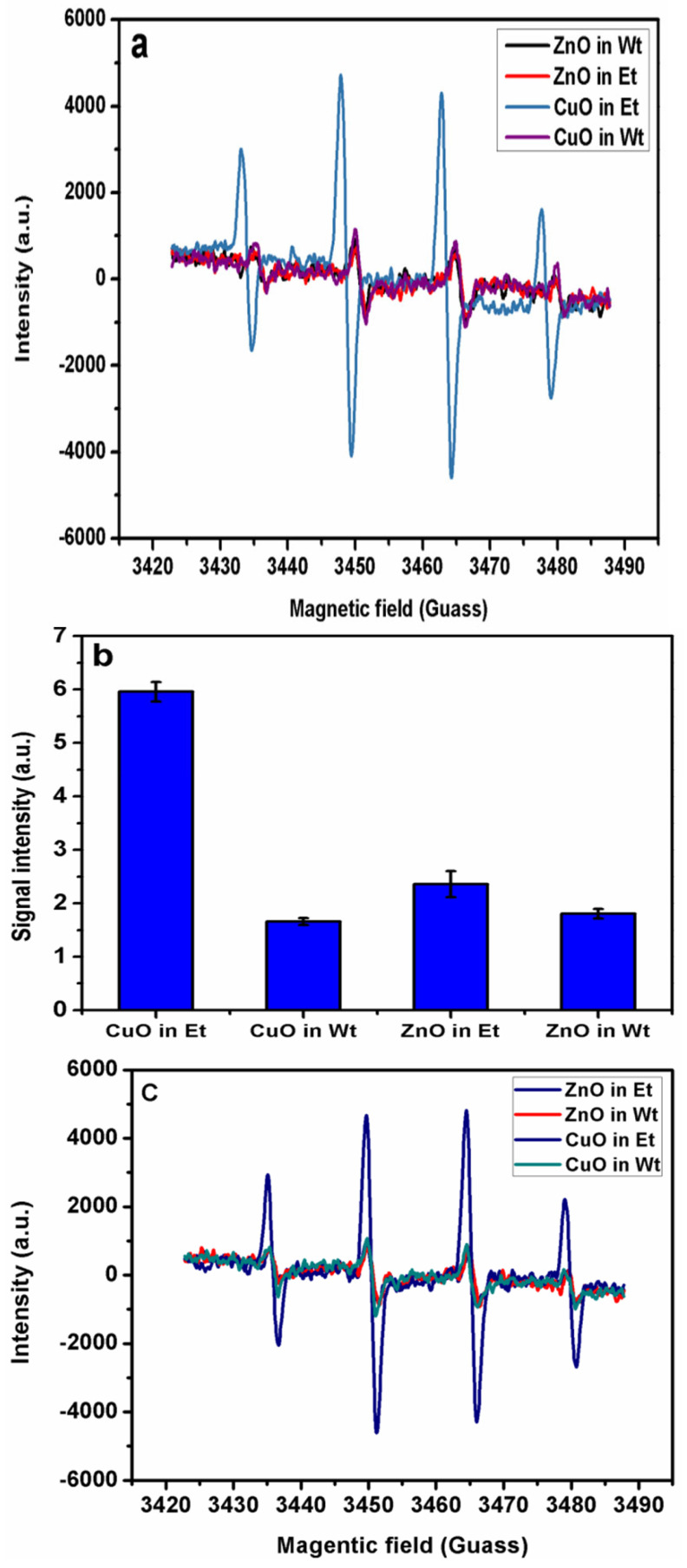
(**a**) ROS generation by CuO and ZnO NPs in either water or ethanol (Et) in presence of the spin trap DMPO alone; (**b**) Integrated area of DMPO-OH spin adduct generated from CuO and ZnO NPs synthesized in either water and ethanol (Et) solvents; (**c**) ROS formation in a suspension of CuO and ZnO NPs in the presence of DMPO and DMSO.

**Figure 10 bioengineering-11-01234-f010:**
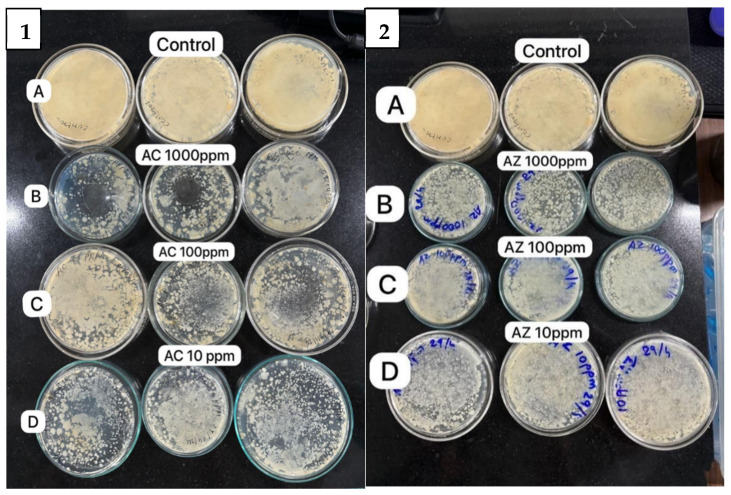
Antibacterial activity of (**1**) Ag-modified CuO nanoparticles (AC) (**A**) control, and at different concentrations (in ppm) of AC NPs (**B**) 1000, (**C**) 100, (**D**) 10; and (**2**) Ag-modified ZnO nanoparticles (AZ) (**A**) control, and at different concentrations (in ppm) of AZ NPs (**B**) 1000, (**C**) 100, (**D**) 10; against soil bacteria.

**Figure 11 bioengineering-11-01234-f011:**
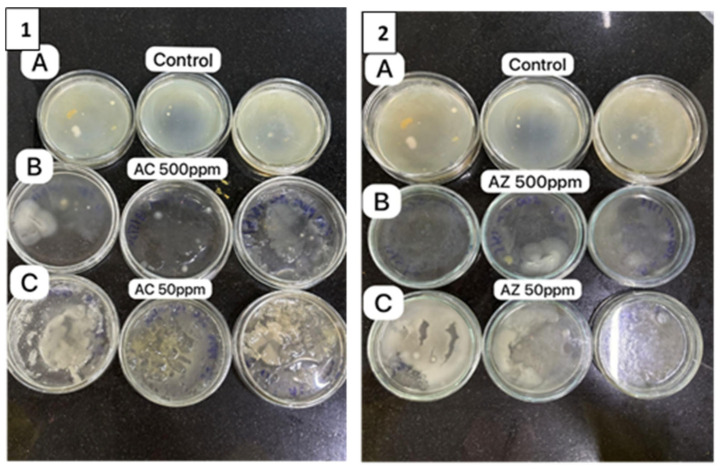
Antimicrobial activity of (**1**) Ag-modified CuO nanoparticles (AC), (**A**) control, and at different concentrations (in ppm) of AC NPs (**B**) 500, (**C**) 50, and (**2**) Ag-modified ZnO nanoparticles (AZ) (**A**) control, and at different concentrations (in ppm) of AZ NPs (**B**) 500, (**C**) 50, against air bacteria.

**Figure 12 bioengineering-11-01234-f012:**
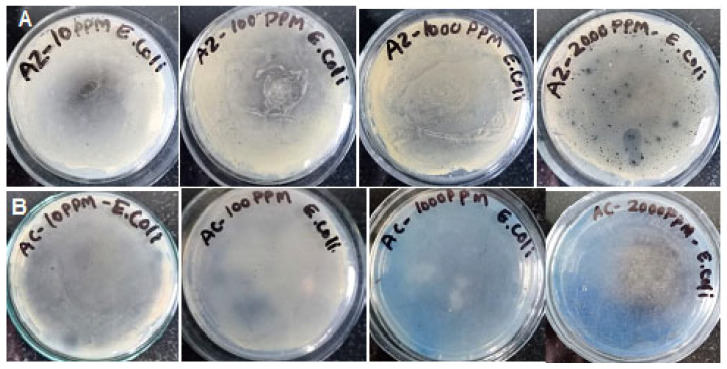
Zone of inhibition (ZoI) test to prove the antibacterial activity of (**A**) AZ and (**B**) AC NPs against *E. coli.*—effect of the concentration of the NPs (100, 1000, 2000 ppm).

## Data Availability

The authors are willing to share any data related to the study through a reasonable request via an email.

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
