# Peer review of "Antibacterial Activity of Silver-Modified CuO Nanoparticle-Coated Masks"

_bioengineering, 2024, doi:10.3390/bioengineering11121234_

Round 1

Reviewer 1 Report

Comments and Suggestions for Authors

See attached referee report

Author Response

Referee 1: 

Recommendation: Reject 
This is a somewhat interesting, but very confusing paper. 

Comment 1: Section 1 :  The whole study is antibacterial—so the virus discussion of SARS-Covid is irrelelvant. The title should also probably be changed from ‘Combatting microbes with…’ to ‘Antibacterial activity of….’ 

Response 1: : Authors agree with the comment, and accordingly title of the paper has been changed as shown on lines 2-3 in the revised paper as shown in track changes and highlighting mode. 
Moreover, reason for the mentioning of viruses has been clarified on lines 40-41 and onlines 47-50.

Comment 2: Section 2.1 : It is not stated what base face mask was coated. Presumably, the surface characteristics of this material (e.g. polypropylene, polyethylene, cellulose) is important. 

Response 2: We thank the referee for the useful advise. The suggested information is provided on lines 104-107 in the revised paper. The change was shown in track changes mode.

Comment 3: What is the procedure for collecting the NPs were collected from the dried coated face masks?
Response 3: We thank the referee for the careful evaluation of our work and letting us fill the missing gaps. The suggested procedure was added to the revised paper as shown on lines 115-118 .

Comment 4: None of the experiments was conducted on the coated facemasks-rather they were performed on the NPs-the face mask appears only as the substrate on which the colloidal NPs are deposited. so the face mask is not relevant to the conclusions. Shouldn’t there also be a control with just Ag NPs? Sections 2.4.1 & 2.4.2 These are uncontrolled exposures; the bacteria are not characterized.

Response 4: The referee was correct in his understanding that XRD and FT-IR analysis were done on the Ag modified CuO, Ag modified ZnO extracted from the coated masks. However, the FE-SEM studies were performed on the real samples, namely, nanoparticle coated masks. Such details can be found on lines 183-192 in the revised paper. The revision made to the paper following the comment of the referee can be found on lines 420-434.

Comment 5: Section 3.1 From the TEM images, there should be an estimate of the size of the NPs-this is missing. Is there a difference in particle size between CuO and ZnO, and also when doped with Ag? 
Response 5: We understood that the referee was pointing to the FE-SEM studies shown on lines 190-192.   Follwing the comment of the referee, thed discussion on the relative average particle sized of Ag modified CuO and Ag modified ZnO is now added as shown on lines 186-190.

Comment 6: Section 3.2 From the XRD, there should be an identification of crystal structure (the peaks are very sharp). As I read the discussion (following Figure 4), CuO is identified as monoclinic, but this changes to cubic when doped with Ag; this would argue against surface coating. The situation appears to be different with ZnO (discussion following Figure 5); the bare material is Wurtzite, with cubic Ag phase—is the Ag coating thick enough to support a crystal structure? 

Response 6:  We thank the referee for the useful insight on doped metal oxides and their crystal structure.  As discussed in lines 201- 244, the Ag-CuO or the Ag-ZnO systems are not Ag doped metal oxides.  The individual phases of monoclinic CuO and cubic Ag are present in the Ag modified system and  like wise individual phases of cubic Ag and Wurtzite phase of ZnO were present in the Ag ZnO system.  

Comment 7: Section 3.3 : I don’t understand what the FT-IR informs about the NPs. The red and black curves in Figures 6 & 7 look very similar, so it is not clear what changes with Ag-coating. 

Reponse 7: We thank the referee for the critical evaluation. Indeed the referee is correct. The difference between FT-IR spectra of bare ZnO and Ag modified ZnO is not perceptible.  The change is clarified on lines 265-267 and lines 278 and 282.   Moreover  Figure 6, the correct image of the  FT-IR figure of Ag modified CuO is provided in the revised paper. We are sorry for the inconvenience that caused the confusion.

Comment 8: Section 3.4  The ESR results are probably the most important evidence to interpret the antibacterial efficacy. 
Response 8: We thank the referee for rightly pointing out the correlation between the reactive oxygen species concentration evident from EPR signal intensity and its relationship with the antibacterial activity. Such an explanation is provided in detail in lines 302-326. 

Comment 9: In Table 1, I quibble with sloppy use of significant figures: it doesn’t make sense to have the 5000 ppm column list results 4 (one significant figure—zero significant figures past the decimal point) with an error of 0.08 (2 decimal places past the decimal point). Also, how is the error calculated? The control is without any NP dosing of the agar plate—so this control column is trivial (definition of zone of inhibition). It would be more interesting if the other 3 NPs were also studied. Concentrations other than 5000 ppm would also be interesting to study. 
Response 9:  We thank the referee for the useful suggestion. Following the comment the correction has been made as shown in lines 336-350.

Comment 10: My previous criticism of the oil and air exposure pertains to Figures 9 and 10. The authors need to extract more from this study than just the anecdotal images.  Figure 11 supports the zone inhibition study of Table 1. These should be placed together with evidence of all 4 NPs.
Response 10:  Following the comments of the referee, the revision has been made to the paper as shown on lines 418-435.

Comment 11:  Author Contributions 
This has not been filled out—we just see the journal template (sloppy) 
Response 11:  Following the referee’s comment the revision has been made as shown on lines 453-455 in the track changes mode. 
Comment 12:  Data Availability Statement 
This has not been filled out—should probably be Not applicable. 
Response 12:  Following the advice of the referee, the corresponding addition was made as shown on lines 458-459. 

Comment 13: 
Conflicts of Interest 
This has not been filled out—we just see the journal template (sloppy)
Response 13:  We thanks the referee for drawing our attention to the journal template.  The suggested details are now provided in the journal template as shown on lines 468. 

Reviewer 2 Report

Comments and Suggestions for Authors

The manuscript reports the coating of surgical mask with AgCuO and AgZno nanoparticles and the study of their antibacterial activity. My comments are following  

1)has the stability of the coatings been assessed under experimental conditions simulating the actual application of the mask?

2) the coating load should be reported

3) the experimental conditions of the coating process should be described in more detail.

4) in the XRD spectrum of AgCuO, CuO peaks should be signed to support the sentence

5) some typing errors are present, please check the entire manuscript carefully (i.e. lines 190, 192, 195)

6) line 196-197: in this sentence the planes (111) and (220) refer to which crystal structure?

7) It seems to me that FTIR analysis is not very indicative for the formation of metal oxide nanoparticles. The authors write that the signal at 588cm-1 is indicative for the formation of metal oxide nanoparticles. the FTIR spectrum of the metal precursors and solvents should be reported to confirm this assignation.

Author Response

Referee 2:

Comments and Suggestions for Authors

The manuscript reports the coating of surgical mask with Ag-CuO and Ag-ZnO nanoparticles and the study of their antibacterial activity. My comments

are following

Comment 1: Has the stability of the coatings been assessed under experimental

conditions simulating the actual application of the mask?

Response 1:   We thank the referee for the useful comment. Following the referee’s comment, the revision made to the paper is shown on lines 436-440.

Comment 2:  The coating load should be reported.

Response 2: We thank the referee for the useful insight. Folowing the comment the paper has been revised and the details of the loading was provided on lines 190-191.

Comment 3: The experimental conditions of the coating process should be described in more detail.

Respones 3: Experimental conditions have been described in detail as shown on lines 104 – 106 and lines 115-118 in the revision.

Comment 4: In the XRD spectrum of AgCuO, CuO peaks should be signed to support the sentence.

Respone 4:  The description of the observed XRD pattern of CuO and Ag modified CuO has been revised as shown on lines 202-245.

Comment 5: some typing errors are present, please check the entire manuscript carefully (i.e. lines 190, 192, 195)

Response 5: The paper has been completely revised following the comments of the referee and the changes made are shown in track changes mode throughout the revised manuscript.

Comment 6: line 196-197: in this sentence the planes (111) and (220) refer to which crystal structure?

Response 6:  Following the referees comment, explantion on the cubic crystal structure of the Ag in both Ag modified CuO and Ag modified ZnO is shown on lines 133-236.

Comment 7: It seems to me that FT-IR analysis is not very indicative for the formation of metal oxide nanoparticles. The authors write that the signal at 588cm-1 is indicative for the formation of metal oxide nanoparticles. the FT-IR spectrum of the metal precursors and solvents should be reported to confirm this assignment.

Response 7: Stretching frequency at 588 cm-1 indicates the formation of metal oxide bond.  Our previous study published in ACS Nano indicates the formation of Fe-O bond at 575 cm-1 .  We fully agree with the comment that the FT-IR spectra of the precursor or the solvent may offer a better view.  However, we observed a concurrence in the XRD and FT-IR result and no trace of precursor was observed in the XRD analysis.

Reviewer 3 Report

Comments and Suggestions for Authors

This work deals with the deposition of Ag-ZnO and Ag-CuO NPs on facemasks to improve their barrier effect against bacteria and viruses. The antibacterial properties of ZnO, CuO and Ag have been known for a long time. The fact that CuO performs better than ZnO is not new either. The idea of covering a facemask worn directly over the airways with NPs poses a serious problem in terms of the risk of inhaling these NPs (see WHO recommendations on this subject: https: //iris.who.int/bitstream/handle/10665/259671/9789241550048-eng.pdf). The authors should explain how this risk could be addressed. In addition, the ineffectiveness of the mask, which the authors use as an argument, is mainly due to the way in which it is worn and leaks through contact with the wearer's skin.

Other technical problems should also be mentioned:

1- The authors claim in the abstract that the ZnO and CuO NPs are decorated with Ag. The SEM images in Figure 2 show nothing of the sort.

2- The intensity of the XRD patterns in Figures 3, 4 and 5 must be in arbitrary units. In Figure 3, the peak at 2q = 32.52° for CuO corresponds to the (110) plane and not (100) as stated in the text. For the Ag-CuO patterns, there appears to be a discrepancy between the positions of the peaks in the Ag-CuO spectrum and the values given in the text. The scale of Figures 4 and 5 must be identical to facilitate comparison.

3- The FTIR spectra in Figures 6 and 7 show no difference due to the presence of Ag, whereas in the text the authors state: "FT-IR spectroscopy analysis proved the synthesis of Ag-CuO NPs in ethanol-201 water by sonication method".

Author Response

Referee 3:

Comment 1: This work deals with the deposition of Ag-ZnO and Ag-CuO NPs on face-masks to improve their barrier effect against bacteria and viruses. The antibacterial properties of ZnO, CuO and Ag have been known for a long time. The fact that CuO performs better than ZnO is not new either. The idea of covering a facemask worn directly over the airways with NPs poses a serious problem in terms of the risk of inhaling these NPs (see WHO recommendations on this subject: https: //iris.who.int/bitstream/handle/10665/259671/9789241550048-eng.pdf). The authors should explain how this risk could be addressed. In addition, the ineffectiveness of the mask, which the authors use as an argument, is mainly due to the way in which it is worn and leaks through contact

with the wearer's skin.

Response 1: We thank the referee for the critical evaluation of the potential of the product, namely the NP coated masks.  Following the referee’s comments, the paper has been revised as shown on lines 437-441.

Other technical problems should also be mentioned:

Comment 2:  The authors claim in the abstract that the ZnO and CuO NPs are decorated with Ag. The SEM images in Figure 2 show nothing of the sort.

Response 2: We fully agree with the insight of the learned referee that the FE-SEM did not include EDXA to show the presence of both Ag and Cu or Ag and Zn.  However the XRD analysis proved beyond doubt that the systems we have is Ag modified metal oxides (CuO/ZnO) as discussed in details in lines  202-245.

Comment 3:  The intensity of the XRD patterns in Figures 3, 4 and 5 must be in arbitrary units. In Figure 3, the peak at 2θ = 32.52° for CuO corresponds to the (110) plane and not (100) as stated in the text. For the Ag-CuO patterns, there appears to be a discrepancy between the positions of the peaks in the Ag-CuO spectrum and the values given in the text. The scale of Figures 4 and 5 must be identical to facilitate comparison.

Response 3: We thank the referee for the valuable correction. We are sorry for the mistake. Folloring the referee’s comments the revision has been made as shown on lines 209, 213, 243 and 247.

Comment 4:  The FT-IR spectra in Figures 6 and 7 show no difference due to the presence of Ag, whereas in the text the authors state: "FT-IR spectroscopy analysis proved the synthesis of Ag-CuO NPs in ethanol water by sonication method".
Response 4:  We thank the referee for pointing out our mistake. Infact, Figure 6 and 7 were the same which is a mistake. Now the correct figures have been provided and the text has been revised as shown on  lines 269 and 286.

Round 2

Reviewer 1 Report

Comments and Suggestions for Authors

Please see attached Referee Comments on the Author Responses

Author Response

Referee 1:

Recommendation:       Major revisions still needed

Comment 1: Section 1 : The whole study is antibacterial—so the virus discussion of SARS-Covid is irrelelvant. The title should also probably be changed from ‘Combatting microbes with…’ to ‘Antibacterial activity of….’

Response 1: Authors agree with the comment, and accordingly title of the paper has been changed as shown on lines 2-3 in the revised paper as shown in track changes and highlighting mode.

Moreover, reason for the mentioning of viruses has been clarified on lines 40-41 and onlines 47-50.

Referee Comment R1:  I would still eliminate the first paragraph (lines 46-61).  In the second paragraph, I would replace ‘corona virus’ (line 62) with ‘transmissible respiratory diseases’.

Response R1: We thank the referee for the useful advice. Following the referee’s comment the first paragraph (lines 35-49) is deleted.  Moreover, in the second paragraph the term ‘corona virus’  is replaced with ‘transmissible respiratory diseases’ (lines 50-51)

Comment 2: Section 2.1 : It is not stated what base face mask was coated. Presumably, the surface characteristics of this material (e.g. polypropylene, polyethylene, cellulose) is important.

Response 2: We thank the referee for the useful advise. The suggested information is provided on lines 104-107 in the revised paper. The change was shown in track changes mode.

Referee Comment R2:  OK.

Response R2: We thank the referee for accepting our response.

Comment 3: What is the procedure for collecting the NPs were collected from the dried coated face masks?

Response 3: We thank the referee for the careful evaluation of our work and letting us fill the missing gaps. The suggested procedure was added to the revised paper as shown on lines 115-118 .

Referee Comment R3: Good—this clarifies that the NPs are excess material from the coating process.

Response R3: We thank the referee for accepting our response.

Comment 4: None of the experiments was conducted on the coated facemasks-rather they were performed on the NPs-the face mask appears only as the substrate on which the colloidal NPs are deposited. so the face mask is not relevant to the conclusions. Shouldn’t there also be a control with just Ag NPs? Sections 2.4.1 & 2.4.2 These are uncontrolled exposures; the bacteria are not characterized.

Response 4: The referee was correct in his understanding that XRD and FT-IR analysis were done on the Ag modified CuO, Ag modified ZnO extracted from the coated masks. However, the FE-SEM studies were performed on the real samples, namely, nanoparticle coated masks. Such details can be found on lines 183-192 in the revised paper. The revision made to the paper following the comment of the referee can be found on lines 420-434.

Referee Comment R4:

  1. a) I don’t understand the last sentence of the first paragraph (lines 189-190).
  2. b) There is still a lack of characterization of the soil and airborne bacteria. I continue to question the relevance of including this ‘experiment’.

Response R4: The text related to loading on the metal oxides nanoparticles is revised based on the referees comments as shown in 178-180.   Following the question of the relevance of the  studies on the experiments related to soil and airborne bacteria, the paper has been revised as shown on lines  330-339.

Comment 5: Section 3.1 From the TEM images, there should be an estimate of the size of the NPs-this is missing. Is there a difference in particle size between CuO and ZnO, and also when doped with Ag?

Response 5: We understood that the referee was pointing to the FE-SEM studies shown on lines 190-192. Follwing the comment of the referee, thed discussion on the relative average particle sized of Ag modified CuO and Ag modified ZnO is now added as shown on lines 186-190.

Referee Comment R5: OK

Response R5: We thank the referee for accepting our response.

Comment 6: Section 3.2 From the XRD, there should be an identification of crystal structure (the peaks are very sharp). As I read the discussion (following Figure 4), CuO is identified as monoclinic, but this changes to cubic when doped with Ag; this would argue against surface coating. The situation appears to be different with ZnO (discussion following Figure 5); the bare material is Wurtzite, with cubic Ag phase—is the Ag coating thick enough to support a crystal structure?

Response 6: We thank the referee for the useful insight on doped metal oxides and their crystal structure. As discussed in lines 201- 244, the Ag-CuO or the Ag-ZnO systems are not Ag doped metal oxides. The individual phases of monoclinic CuO and cubic Ag are present in the Ag modified system and like wise individual phases of cubic Ag and Wurtzite phase of ZnO were present in the Ag ZnO system.

Referee Comment R6: OK

Response R6: We thank the referee for accepting our response.

Comment 7: Section 3.3 : I don’t understand what the FT-IR informs about the NPs. The red and black curves in Figures 6 & 7 look very similar, so it is not clear what changes with Ag-coating.

Reponse 7: We thank the referee for the critical evaluation. Indeed the referee is correct. The difference between FT-IR spectra of bare ZnO and Ag modified ZnO is not perceptible. The change is clarified on lines 265-267 and lines 278 and 282. Moreover Figure 6, the correct image of the FT-IR figure of Ag modified CuO is provided in the revised paper. We are sorry for the inconvenience that caused the confusion.

Referee Comment R7:

  1. a) Figure 6 (revised) should also contain the FTIR for the uncoated CuO NP in order to make the comparison.
  2. b) The only difference seems to be the appearance of the 588 cm-1 band (for Ag-CuO Figure 6) and the 592 cm-1 band (for Ag-ZnO Figure 7). These bands need to be identified, as the whole case for a separate Ag phase rests on these new bands.

Response R7: The FT-IR spectrum of the CuO (Figure 6) was included and a comparison is made with the spectrum of Ag modified CuO as shown on lines 237-248. Moreover the specific bands typical of Ag-CuO and Ag-ZnO were marked in the correspdonging spectra as shown on lines 243-255 (Figure 7) and 256-271 (Figure 8).

Comment 8: Section 3.4 The ESR results are probably the most important evidence to interpret the antibacterial efficacy.

Response 8: We thank the referee for rightly pointing out the correlation between the reactive oxygen species concentration evident from EPR signal intensity and its relationship with the antibacterial activity. Such an explanation is provided in detail in lines 302-326.

Referee Comment R8: OK

Response R6: We thank the referee for accepting our response.

Comment 9: In Table 1, I quibble with sloppy use of significant figures: it doesn’t make sense to have the 5000 ppm column list results 4 (one significant figure—zero significant figures past the decimal point) with an error of 0.08 (2 decimal places past the decimal point). Also, how is the error calculated? The control is without any NP dosing of the agar plate—so this control column is trivial (definition of zone of inhibition). It would be more interesting if the other 3 NPs were also studied. Concentrations other than 5000 ppm would also be interesting to study.

Response 9: We thank the referee for the useful suggestion. Following the comment the correction has been made as shown in lines 336-350.

Referee Comment R9:

  1. a) I am very skeptical that both zones of inhibition are measured to be 4.00 mm and 3.00 mm (Table 1). I suspect that n replicates were performed for each experiment: ZoI was measured for each replicate, and that 4.00 mm and 3.00 mm are supposed to represent the averages over those n replicates; 0.08 mm and 0.09 mm correspond to the SDs. Please replace with the true averages taken from the experimental data.
  2. b) Other NPs and other concentrations?

Response R9: Following the referees concernes, the paper has been revised as shown on lines134-135 and additional details were provided in the electronic supplementary information.

Comment 10: My previous criticism of the oil and air exposure pertains to Figures 9 and 10. The authors need to extract more from this study than just the anecdotal images. Figure 11 supports the zone inhibition study of Table 1. These should be placed together with evidence of all 4 NPs.

Response 10: Following the comments of the referee, the revision has been made to the paper as shown on lines 418-435.

Referee Comment R10:

  1. a) No—added lines 399-422 further detract from the paper—do not add
  2. b) I still feel that the soil and airborne bacteria study (Figures 9 and 10) are anecdotal and do not materially contribute to this study.

Response R10: Following the coments of the referee the added lines namely, lines 387-404, were now deleted.

Comment 11: Author Contributions

This has not been filled out—we just see the journal template (sloppy)

Response 11: Following the referee’s comment the revision has been made as shown on lines 453-455 in the track changes mode.

Referee Comment R11:  OK

Response R11: We thank the referee for accepting our response.

Comment 12: Data Availability Statement

This has not been filled out—should probably be Not applicable.

Response 12: Following the advice of the referee, the corresponding addition was made as shown on lines 458-459.

Referee Comment R12:  OK

Response R12: We thank the referee for accepting our response.

Comment 13:

Conflicts of Interest

This has not been filled out—we just see the journal template (sloppy)

Response 13: We thanks the referee for drawing our attention to the journal template. The suggested details are now provided in the journal template as shown on lines 468.

Referee Comment R13:  OK

Response R13: We thank the referee for accepting our response.

Reviewer 2 Report

Comments and Suggestions for Authors

Authors revised their manuscript addressing my comments. The manuscript can be accepted for publication. 

Author Response

Referee 2:

Comments and Suggestions for Authors

Authors revised their manuscript addressing my comments. The manuscript can be accepted for publication.

Response: We thank the referee for the encouragement and support in improving the quality of the paper by devoting his valuable time.

Reviewer 3 Report

Comments and Suggestions for Authors

Regarding my main concern about the idea of depositing nanoparticles directly on a mask in direct contact with the respiratory tract and the risks of inhaling these nanoparticles, the only answer I've seen is that the deposition is only on one side of the mask. I found nothing else in lines 437-441 of the new version, as stated in the authors' reply. As far as I'm concerned, this doesn't solve the problem of inhalation risks, which is not even mentioned in the text.

As for the authors' arguments about the presence of Ag in the form of nanoparticles rather than as CuO and ZnO dopants, I think these arguments would be valid if the concentration of Ag were not so low.

Finally, my comment about the same scale for Figures 4 and 5 has not been taken into account.

Author Response

Referee 3:

Comment 1: Regarding my main concern about the idea of depositing nanoparticles directly on a mask in direct contact with the respiratory tract and the risks of inhaling these nanoparticles, the only answer I've seen is that the deposition is only on one side of the mask. I found nothing else in lines 437-441 of the new version, as stated in the authors' reply. As far as I'm concerned, this doesn't solve the problem of inhalation risks, which is not even mentioned in the text.

Response 1:  We thank the author for carefully following our arguments.  The referee is right that one of the explanations we offered is that the coating was only on a single side. The other argument was provided on lines 405-409. We apologize the referee for providing wrong line numbers 437-441 in our previous response and this is our mistake.

Comment 2: As for the authors' arguments about the presence of Ag in the form of nanoparticles rather than as CuO and ZnO dopants, I think these arguments would be valid if the concentration of Ag were not so low.

Response 2:    We thank the referee for correctly understanding the procedures we have doped.  The referee is indeed correct that the concentration of Ag  precursor was  not low as shown on lines 98-99.

Comment 3:  Finally, my comment about the same scale for Figures 4 and 5 has not been taken into account.

Response 3:  We apologize the referee for the misunderstanding. Infact, we shared the revised figures 4 and 5 with the editor soon after submission of the revised file requesting that the files to be passed on to you. Unfortuantely, that has not happen for some reason. Now the revised XRD plots of Figures 4 and 5 are provided in the revised manuscript as shown on lines 223-227 and lines 230-234.

Round 3

Reviewer 1 Report

Comments and Suggestions for Authors

The authors have now adequately addressed my comments 1,4,7,9,10,11.

The paper is now acceptable for publication.

Author Response

Comments of referee 1:  The authors have now adequately addressed my comments 

The paper is now acceptable for publication.

Response to referee 1:  We thank the referee for devoting his valuable time and providing valuable comments and enabling us to revise the paper to a form where it stands now accepted. Thank you sir. 

Reviewer 3 Report

Comments and Suggestions for Authors

Again, the problem is not the stability of the antibacterial properties of these masks over time. The problem is to demonstrate that some of these NPs would not pass through the mask and be inhaled under the effect of the mask wearer's suction during breathing.

The authors should clearly state in their manuscript that this method still requires experimental verification to demonstrate the harmlessness of these NPs coatings in relation to the risks of their inhalation.

Author Response

Comment 1:  Again, the problem is not the stability of the antibacterial properties of these masks over time. The problem is to demonstrate that some of these NPs would not pass through the mask and be inhaled under the effect of the mask wearer's suction during breathing.

The authors should clearly state in their manuscript that this method still requires experimental verification to demonstrate the harmlessness of these NPs coatings in relation to the risks of their inhalation.

Response 1:  We thank the referee for clarifying his concerns.  Following the suggestion the manuscript has been revised as shown on lines 330-332.

Round 4

Reviewer 3 Report

Comments and Suggestions for Authors

The sentence added by the authors in lines 330-332 "However, this method still requires experimental verification to demonstrate the harmlessness of these NPs coatings in relation to the risks of their inhalation." seems to me satisfactory and adds a reasonable note of health caution to their work. However, it would be better placed at the end of the summary and at the end of the conclusion.

Author Response

Comment 1: The sentence added by the authors in lines 330-332 "However, this method still requires experimental verification to demonstrate the harmlessness of these NPs coatings in relation to the risks of their inhalation." seems to me satisfactory and adds a reasonable note of health caution to their work. However, it would be better placed at the end of the summary and at the end of the conclusion.

Response 1: We thank the referee for devoting his time and providing the useful suggestion. Following the advice, the paper has been revised as shown on lines 328-330, 331 and 344-346.
